

# Coral recruits demonstrate thermal resilience

Annika M. Lamb[1,2,3], Lesa M. Peplow[1], Peter L. Harrison[4], Craig A. Humphrey[1], Lorenzo Latini[1], Guy A. McCutchan[1] and Madeleine J. H. van Oppen[1,3]

[1] Australian Institute of Marine Science, Townsville, Queensland, Australia
[2] AIMS@JCU, James Cook University of North Queensland, Townsville, Queensland, Australia
[3] School of Biosciences, The University of Melbourne, Parkville, Victoria, Australia
[4] Marine Ecology Research Centre, Southern Cross University, Lismore, New South Wales, Australia

Corresponding author
Annika M. Lamb,
a.lamb@aims.gov.au

## ABSTRACT

Marine heatwaves are becoming more frequent during summer and pose a significant threat to coral reef ecosystems. Restoration efforts have the potential to support native coral populations and guard them against some degree of environmental change, while global action against climate change takes place. Interspecific hybridization is one approach through which resilient coral stock could be generated for restoration. Here we compared the performance of *Acropora kenti* and *A. loripes* hybrid and purebred coral recruits under a simulated thermal stress event. *A. kenti* eggs were successfully fertilized by *A. loripes* sperm to produce 'KL' hybrids, but no 'LK' hybrids could be produced from *A. loripes* eggs and *A. kenti* sperm. Despite corals in the elevated treatment accruing thermal stress (>12 degree heating weeks over 2 months) known to result in mass bleaching, both purebred and hybrid recruits showed no signs of stress under the simulated temperature regime, based on the performance indicators survivorship, size, color (a proxy of bleaching), and photochemical efficiency of photosystem II. Comparisons between the hybrids and purebreds studied here must be interpreted with caution because hybrid sample sizes were small. The hybrids did not outperform both of their purebred counterparts for any metrics studied here, demonstrating that there are limitations to the extent to which interspecific hybridization may boost the performance of coral stock. In general, the purebred *A. loripes* recruits performed best under both ambient and elevated conditions. The performance of the KL hybrid corals was similar to the maternal parental species, *A. kenti*, or not significantly different to either parental purebred species. The Symbiodiniaceae communities of the KL hybrids were characteristic of their maternal counterparts and may have underpinned the performance differences between the *A. kenti*/KL hybrid and *A. loripes* recruits.

## INTRODUCTION

Extreme marine heatwaves are becoming more frequent and ocean surface temperatures are rising globally at unprecedented rates (*Oliver et al., 2019*; *IPCC, 2022*). Temperature-induced stress can cause the breakdown of the mutualistic relationship between corals and

their photosynthetic endosymbionts in the family Symbiodiniaceae—a process termed coral bleaching that often results in the starvation and death of the coral host (*Hoegh-Guldberg, 1999*). Heatwaves have caused mass bleaching and loss of coral cover across many reef regions globally over recent decades (*Heron et al., 2016*; *Great Barrier Reef Marine Park Authority, 2017*; *Hughes et al., 2017*; *Le Nohaic et al., 2017*). The drastic and rapid decline of coral reefs indicates that the natural tolerance and rate of adaptation of coral may be insufficient to ensure their persistence in a future where summer heatwaves are expected to become the norm (*Pandolfi et al., 2003*; *van Hooidonk et al., 2016*).

Active restoration programs may be required to maintain or replenish the diversity and abundance of corals on some reefs. This can be done using native, purebred coral stock; however, such material contributes little new diversity to populations and can be maladapted to rapidly-changing environmental conditions (*Broadhurst et al., 2008*; *Sgro, Lowe & Hoffmann, 2011*). Instead, the use of coral stock with enhanced resilience may increase the success of restoration (*van Oppen et al., 2015*, *2017*). Resilient corals would ideally survive, grow, and reproduce despite marine heatwaves and rising sea surface temperatures and so would effectively replenish reefs and guard them against climate chance.

Resilient stock may be produced through interspecific hybridization (*Chan, Hoffmann & van Oppen, 2019*). Hybrids contain the gene variants (alleles) of two different species and harbor novel genetic combinations (*Kitchen et al., 2019*). The novel genetic combinations harbored by hybrids can manifest as adaptive traits such as enhanced resilience (*Chan, Hoffmann & van Oppen, 2019*). There is evidence that interspecific hybridization naturally occurs on some reefs and has been evolutionarily significant among certain coral lineages (*van Oppen et al., 2001*; *Fogarty, 2012*; *Kitchen et al., 2020*). Furthermore, a long-term experiment comparing the performance of hybrid and purebred corals under ambient and predicted future climate conditions has demonstrated that, under prolonged elevated temperature and $CO_2$ conditions, hybrids generally grow and survive equally or better than their purebred parental species (*Chan et al., 2018*). Hybrid stocks must also be tested under extreme heatwaves, or simulations of extreme heatwaves, so that the value of interspecific hybridization as an intervention tool can be comprehensively assessed.

Here we assessed the performance of hybrid and purebred coral recruits under a simulated extreme heatwave. We compare the performance of the hybrid and purebred corals under ambient and heatwave conditions by analyzing their survivorship, size, bleaching resistance, and/or photochemical efficiency using a nested design. We hypothesized that the hybrid corals would demonstrate improved performance relative to the purebred corals (hybrid vigor) under heatwave conditions (*Shull, 1908*).

## METHODS

### Coral stock generation and settlement

Sections with approximate diameters of 30 cm of gravid *Acropora kenti* (N = 14; previously referred to as *A. tenuis*; *Bridge et al., 2023*) and *A. loripes* (N = 13) colonies were collected in November 2019 from Davies Reef in the central Great Barrier Reef (−18.82, 147.64; Great

Barrier Reef Marine Park Authority Permit G12-35236.1) which experienced extreme temperatures that resulted in mass bleaching in 2017 and 2020 (post broodstock collection). The colonies were collected prior to summer when the temperature at Davies Reef was recorded at four meters depth as 27.8 °C. The colonies were held in flow-through systems in the National Sea Simulator at the Australian Institute of Marine Science (AIMS) at 27.8 °C. These species were selected because they can be successfully cross-fertilized using *in vitro* fertilization to produce viable hybrids for which there is an existing knowledge base (*Chan et al., 2018*, *2019*; *Chan, Peplow & van Oppen, 2019*). *Acropora kenti* and *A. loripes* typically spawn approximately 2 h apart and so it is expected that natural interspecific hybridization is limited between them in the wild (*Baird et al., 2021*).

The colonies were watched from sunset until approximately 23:00 for the 10 days following the full moon (12th of November) and their spawning activity was recorded. Colonies showing signs of 'setting', indicating imminent spawning (*Babcock et al., 1986*), were isolated and once gametes were released, their egg-sperm bundles were collected and separated into eggs and sperm using a 100 µm mesh filter. The density of the sperm collected from each colony was calculated using four replicate counts on a hemocytometer and adjusted to $1 \times 10^7$ sperm per mL. The sperm of all colonies from the same species were then combined in equal parts creating a mixed sperm solution for both *A. kenti* and *A. loripes*. Crosses were then conducted to produce two purebred and two hybrid offspring groups: *A. kenti* purebreds, *A. loripes* purebreds, hybrids resulting from fertilization of the eggs of *A. kenti* by the sperm of *A. loripes* (KL hybrids), and hybrids resulting from fertilization of the eggs of *A. loripes* by the sperm of *A. kenti* (hereafter referred to as LK hybrids). The mixed sperm solution was added to the eggs of each conspecific colony at a density of $1 \times 10^6$ sperm per mL to optimize fertilization rates (*Willis et al., 1997*) and generate the purebreds, and to the eggs of each colony of the other species to generate the hybrids. The eggs of each colony were fertilized in a separate reaction (rather than all eggs being combined in the same reaction) to minimize the risk of cross-contamination of the hybrid crosses by conspecific fertilization of the eggs of one colony by residual sperm on the eggs of a second colony. Post-fertilization, the embryos from successful crosses were combined according to offspring group.

The coral larvae were maintained in ~85 L larval rearing tanks with air and water flow for approximately one week until they were competent to settle. Once competent, planula larvae were introduced into 50 L acrylic tanks containing 15–20 biologically conditioned $100 \times 100$ mm terracotta tiles at an approximate density of 200 larvae per tile. Larvae from the different offspring groups were settled separately so that each tile contained corals of just one offspring group. The tiles had been biologically conditioned in coral rearing tanks for 6 weeks prior to use. Upon settlement, the larvae were exposed to Symbiodiniaceae that had been isolated from tissue fragments from adult *A. kenti* colonies (with uncharacterized Symbiodiniaceae communities) from the reefs fringing Yunbenun (Magnetic Island; −19.14, 146.82) following *Chan et al. (2018)*. It was planned that the recruits would be deployed at Yunbenun as a component of a field-based experiment, but COVID pandemic restrictions prevented this from happening. The recruits were inoculated with Symbiodiniaceae from the intended recipient environment and not Davies Reef (parental

reef) to eliminate any risks posed to the recipient site by translocating Symbiodiniaceae. Instead, post-settlement, the coral recruits were reared in 500 L tanks in outdoor systems at AIMS under natural light at 27.8 °C for 5 months. Sediment and algal biomass were cleaned from the holding systems at least once a week, and more regularly when required.

## Experimental conditions

The terracotta tiles with live 5-month-old coral recruits were distributed randomly among eight 200 L tanks with one face up; the corals that settled on this face of the tiles became the experimental subjects. The flow in the systems was such that the seawater volume was replaced every 2 hours. The tanks were illuminated using AquaIllumination Sol LED lights with a 12-h light/dark cycle, a 1 h ramp up and ramp down period on either end of the day, and approximately 100 PAR of light reached the surface of the tiles. The corals were fed 0.5 nauplii/mL of artemia and 2,000 cells/mL of a mixed-species microalgae solution daily (*Conlan et al., 2019*). The tanks were cleaned at least twice a week, and more regularly if required. The corals were held in their respective tanks for a two-week period allowing for acclimation of the corals to the rearing conditions at 27.8 °C. The average daily temperature of the four ambient tanks was maintained at 27.8 °C, the long-term average water temperature at Davies Reef from November to March, as recorded by the Davies Reef Weather Station (DRWS) operated and maintained by AIMS, for the duration of the experiment (Fig. 1). After the acclimation period, the temperature in four of the tanks was increased by 0.1 °C per day for 9 days until it equaled the average daily temperature just prior to the 2020 summer heatwave recorded at the DRWS that resulted in mass bleaching on the Great Barrier Reef (GBR) (Fig. 1). The temperature in the elevated temperature treatment tanks aligned with the 2020 summer heatwave from days 9–31 of the experiment until the peak of the heatwave had been reached (Fig. 1). The elevated tanks were then held at an elevated temperature for 5 weeks after the peak was reached (Fig. 1). This equated to the corals in the elevated treatment being exposed to a total of 12.6 cumulative degree heating weeks (DHWs), as calculated based on the long term Davies Reef average water temperatures and using the NOAA Coral Reef Watch Bleaching Alert System values (Fig. 1; *National Oceanic and Atmospheric Administration Coral Reef Watch, 2021*). The average daily water temperatures of the elevated treatment tanks ranged from 27.9–30.6 °C. Oscillating fluctuations of 0.3 °C on either side of the daily peak were maintained in both the ambient and elevated treatments to mimic daily cycling of temperatures.

## Photo analyses

All tiles were imaged using a high-resolution camera (Nikon D810) with a scale bar and the D-side of the Coral Watch Coral Health Chart (*Siebeck, Logan & Marshall, 2008*) for survivorship, size, and color measurements (details below) at:

1) T2: after 2–3 days of temperature ramping in the elevated treatment.
2) T30: after 30–31 days of temperature ramping when the maximum temperature had been reached in the elevated treatment.

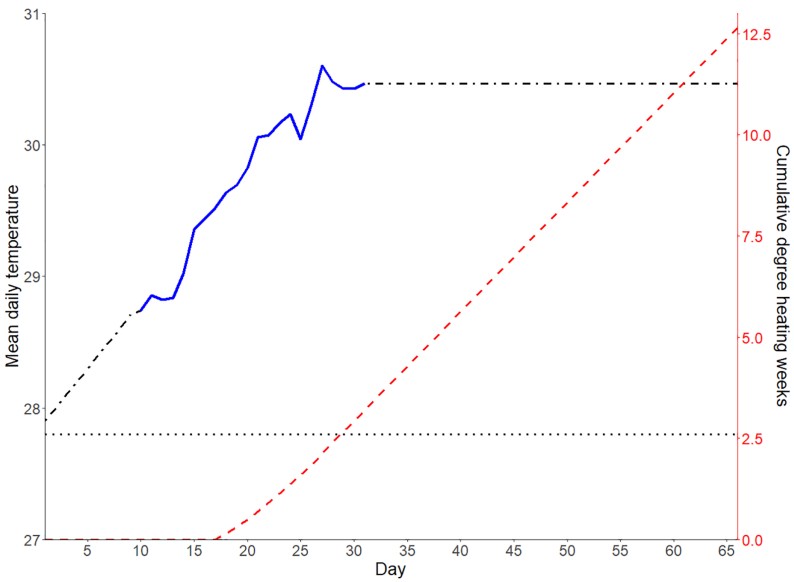

**Figure 1 Schematic of the experimental temperature profiles.** Temperature profiles of the elevated (dash-dot black and continuous blue line) and ambient (dotted black line) experimental treatments. The temperature in the elevated treatment tanks was ramped between days 0–9 of the experiment and simulated the temperatures recorded at Davies Reef during the 2020 mass bleaching event between days 9–31 (continuous blue) of the experiment. The cumulative degree heating weeks (DHWs) experienced by the corals in the elevated temperature treatment over the course of the experiment are also shown (dashed red). After 31 days, the elevated treatment tanks were held at the peak of the heat wave until the corals had experienced >12 degree heating weeks (DHWs).

3) T65: after 65–66 days of the temperature treatment when the corals in the elevated treatment had experienced a total of 12.6 DHWs.

The images were analyzed using ImageJ software (*Schneider, Rasband & Eliceiri, 2012*). Recruits that settled in clusters or that grew into one another over the course of the experiment were excluded from further analyses since they could not be considered independent replicates. Corals that grew from the face and onto edges of the tiles were also excluded. Recruits were scored as alive or dead from the images. The images were scaled using the scale bar that was included in the imaging stage to ensure accurate size measurements. Recruits were circled using the freehand selection tool in ImageJ and the total recruit area ($mm^2$) was measured. For color measurements, the images were converted to 8-bit and calibrated using a linear function estimated from the D-side of the Coral Watch Coral Health Chart (*Siebeck, Logan & Marshall, 2008*). From the calibrated 8-bit images, the mean color score over the total area of each recruit was measured. Color score is used here as a proxy for the density of algal symbionts in the coral tissue where a lower number/lighter color indicates a lower algal symbiont density that is indicative of coral bleaching (*Siebeck, Logan & Marshall, 2008*).

## Photochemical efficiency measurements

Dark-adapted maximum photosystem II quantum yield, $F_v/F_m$ (photochemical efficiency), was measured after the acclimation period and every 2 weeks thereafter

until day 56 of the experiment. The $F_v/F_m$ is indicative of the maximum efficiency of PSII and was used to track photosystem health throughout the course of the experiment. The tiles were dark-adapted for 15 minutes prior to the photochemical efficiency of the recruits being measured using an Imaging Pulse Amplitude Modulation (iPAM; Walz) fluorometer with the software ImagingWin (v2.40 b), a Measuring Intensity of three, and Gain of one. The photochemical efficiency of each recruit on the same tile was measured simultaneously.

## Statistical analyses

Statistical analyses and graphics were conducted and produced using RStudio (*RStudio Team, 2016*) with R 4.0.5 (*R Core Team, 2021*). Statistical models were built to test the effects of the fixed variables – temperature treatment (elevated or ambient), offspring group (*A. kenti* purebred, *A. loripes* purebred, and KL hybrid), and time point – on the performance metrics measured over time: survivorship, size, color, and $F_v/F_m$. Survivorship was compared among offspring groups using generalized linear mixed effects models (GLMM) built using the lme4 package (*Bates et al., 2014*). The color and photochemical efficiency of the offspring groups were compared using linear mixed effects models (LMM) built using the lme function in the lme4 package and the maximum likelihood method (*Bates et al., 2014*). Because some recruits grew and others did not grow during the experiment, variance in recruit area increased throughout the experiment. The variance around recruit area was also heteroscedastic among offspring groups. The nlme function from the nlme package (*Pinheiro et al., 2021*) was used to construct LMMs which allowed variance to differ amongst time points and offspring groups and were used to compare the size of the offspring groups. For each performance metric, five models were built that treated tile nested within tank as a random effect:

1) Model one included interaction terms amongst the three fixed variables.
2) Model two included no interaction terms amongst any of the fixed variables.
3) Model three included an interaction between offspring group and treatment.
4) Model four included an interaction between offspring group and time point.
5) Model five included an interaction between treatment and time point.

The performance of the models pertaining to a metric was then compared using likelihood ratio tests. The most complex model with interactions between each of the fixed variables (1 above) was compared to each of the less complex models (2–5 above). Each of the models with one interaction term between a pair of the fixed variables (3–5 above) was also compared to the simplest model which included only the main effects and no interaction terms (2 above). The simplest model (fewest interaction terms) that no other model outperformed (as determined through likelihood ratios) was considered the best fit for the data, reported, and interpreted. *Post-hoc* Tukey's tests were conducted on the best-performing models to compare the performance of the three offspring groups based on each metric.

## Symbiodiniaceae symbiont communities

After 66 days of the heatwave treatment, at the completion of the experiment, tissue was sampled from the remaining corals into ethanol so that the Symbiodiniaceae communities of the corals could be characterized. DNA was extracted from the samples following *Damjanovic et al. (2019)* and the ITS2 region was amplified from the extracted DNA using the SYM_VAR_5.8S2/SYM_VAR_REV primers (*Hume et al., 2013*, *2015*) and protocol outlined in *Hume et al. (2018)*. Triplicate PCR reactions were conducted and then pooled for each sample. Library preparation was conducted on the PCR products following *Maire, Blackall & van Oppen (2021)*. The amplified ITS2 products were sequenced using a paired-end (2 × 250 bp) approach at the Walter and Eliza Hall Institute of Medical Research (Melbourne, Australia) and one MiSeq V3 system (Illumina, San Diego, CA, USA). The SymPortal analytical framework was used to analyze the sequences (*Hume et al., 2019*). SymPortal characterizes ITS2 profiles of within-sample ITS2 defining intragenomic variants (DIVs) that are putatively representative of Symbiodiniaceae taxa. The SymPortal framework was used to conduct standard quality control and analyses of the raw sequencing data (*Hume et al., 2019*). R was subsequently used to analyze the SymPortal quality controlled ITS2 sequences and ITS2 profile abundance data. Samples with less than 1,000 reads were removed from the dataset. The relative abundance of the ITS2 profiles and sequences present in each retained sample were plotted. A pairwise similarity matrix was calculated amongst the sequences using a kmer-based approach and a *k* size of seven (*Fujise et al., 2021*) in the kmer package (*Wilkinson, 2018*). A hierarchical clustering approach was then used to construct a dendrogram from the similarity matrix using the upgma function from the phangorn package (*Schliep, 2011*). The presence, abundance, and relatedness of the ITS2 sequences present in each sample and the dendrogram were then used to calculate weighted unique fraction metric (UniFrac) distances amongst the samples (*Lozupone & Knight, 2005*). Hierarchical clustering of the samples was conducted using these distances and visualized by building a tree using the ggtree package (*Yu et al., 2017*), in which samples were colored by offspring group. Two permutational multivariate analysis of variance (PERMANOVA) models were constructed to test for the effect of offspring group and treatment on UniFrac distance, whilst accounting for the nesting within tanks using the package Vegan (*Oksanen et al., 2020*).

# RESULTS

## Spawning activity and gamete compatibility

The gamete crosses were conducted over two nights. The *A. kenti* colonies began spawning from 19:20 to 19:30 and the *A. loripes* colonies began spawning from 21:30 to 21:35. On the 18th of November, six *A. kenti* and two *A. loripes* colonies spawned and were crossed. On the 19th of November, seven *A. kenti* and four *A. loripes* colonies spawned and were crossed. Some colonies spawned and were crossed on both nights, such that the gametes of a total of nine *A. kenti* and five *A. loripes* colonies were used in this experiment. All purebred crosses were successful. None of the hybrid crosses where the eggs of *A. loripes*

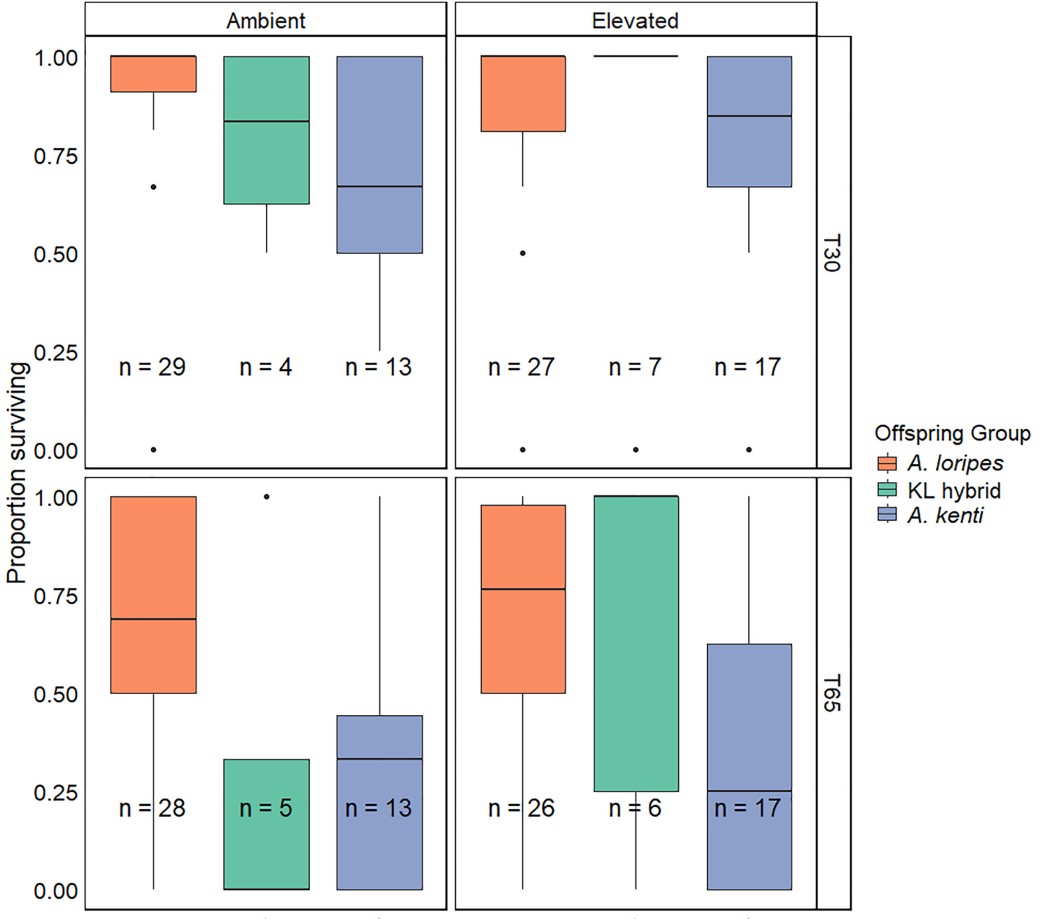

**Figure 2 Juvenile survivorship.** Boxplots depicting the distributions of proportions of surviving recruits per tile for each of the offspring groups (*A. loripes* purebred–orange, KL hybrid–green, and *A. kenti* purebred–blue), in each of the treatments (ambient and elevated), and at timepoints T30 and T65 of the experiment. The horizontal lines of the boxes represent the lower quartile, median, and upper quartile values, the "whiskers" represent the extreme values and dots represent single outlier datapoints. Sample sizes (number of tiles) are shown. There was no significant effect of temperature treatment on survivorship based on generalized linear mixed effects modelling.

were mixed with the sperm of *A. kenti* were successful. The eggs of seven of the nine *A. kenti* colonies were successfully fertilized by *A. loripes* sperm.

## Survivorship

Survivorship of the corals declined between T30 and T65 ($Z = -9.76$, $P < 0.001$; Fig. 2). Temperature treatment had no effect on survivorship ($Z = 0.42$, $P = 0.678$). *A. kenti* purebred recruits had lower survivorship than the *A. loripes* purebred recruits ($Z = -5.25$, $P < 0.001$; Fig. 2). No significant difference in survivorship was detected between the KL hybrids and the *A. kenti* ($Z = 0.84$, $P = 0.668$) or *A. loripes* ($Z = -1.96$, $P = 0.114$) purebred offspring groups (Fig. 2), and the survivorship of the hybrids therefore appeared intermediate compared to the purebreds. None of the GLMMs including interaction terms

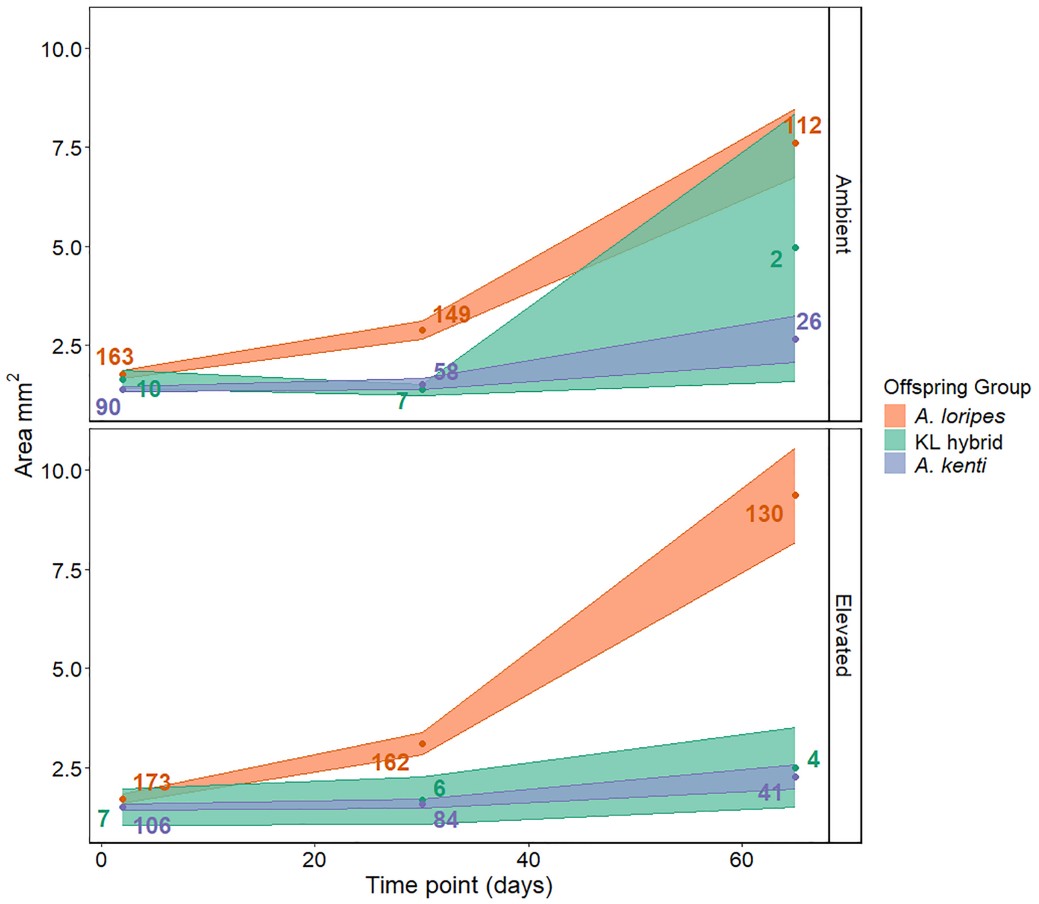

**Figure 3 Juvenile size.** Recruit surface area (*i.e.*, size) over time shown for each of the offspring groups (*A. loripes* purebred–orange, KL hybrid–green, and *A. kenti* purebred–blue) in each of the treatments (ambient and elevated temperature). The data points represent the mean area (mm$^2$), and the upper and lower limits of each ribbon represent the standard error around the mean. Sample sizes (number of recruits) are included next to the data points. There was no significant effect of temperature treatment on size based on linear mixed effects modelling.

amongst offspring group, temperature treatment, or time point were a significantly better fit for the survivorship data than the simplest GLMM that did not include any interactions between these fixed effects (Table S2). The results of the simplest model have therefore been presented.

## Size

The recruits grew throughout the experiment (T = 9.35, $P < 0.001$; Fig. 3) and their growth was not affected by the temperature treatment (T = 0.30, $P = 0.776$). The *A. loripes* purebred corals showed a greater increase in area compared to the KL hybrid corals (2.86-fold difference between *A. loripes* purebreds and KL hybrids in change in mean area from T30 to T65; T = 4.34, $P = 0.001$) and the *A. kenti* purebred corals (5.06-fold difference between *A. loripes* purebreds and *A. kenti* purebreds in change in mean area from T30 to T65; T = 5.93, $P < 0.001$) over time (Fig. 3). There was no significant difference in area over time between the *A. kenti* purebred and KL hybrid corals (T = 0.48, $P = 0.883$; Fig. 3). The

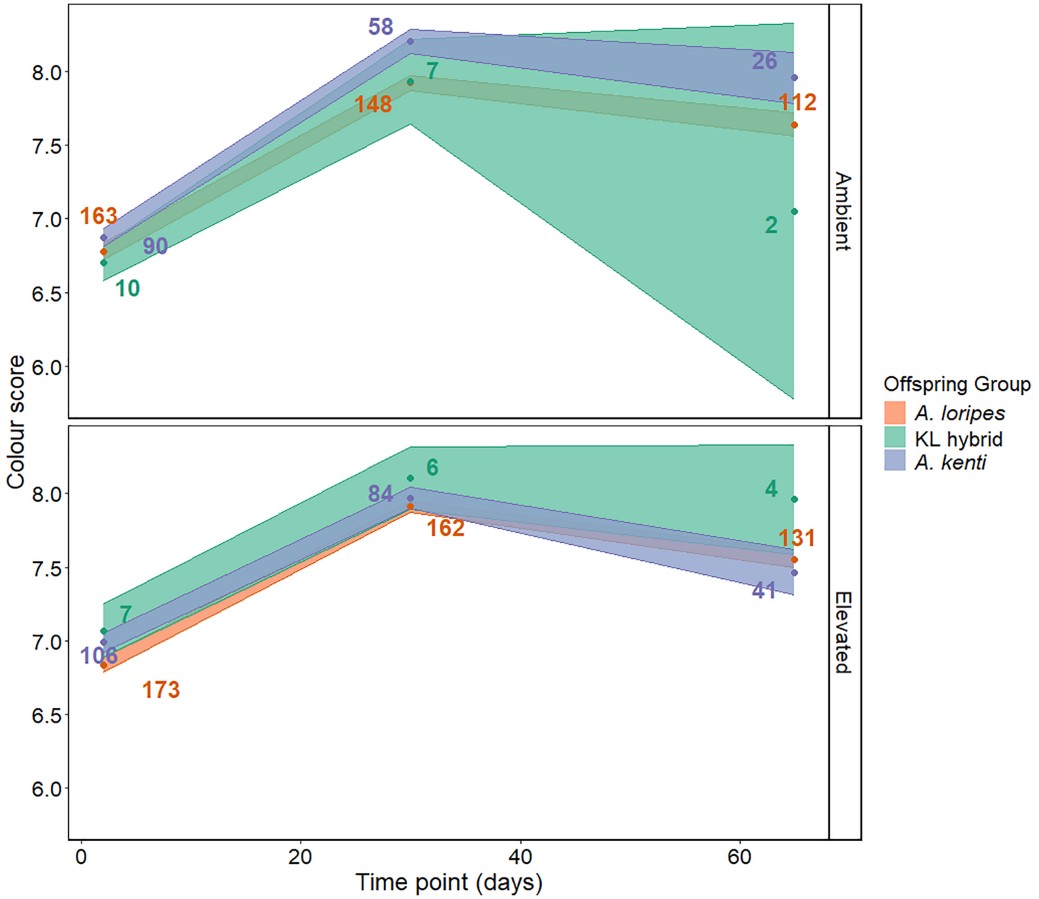

**Figure 4 Juvenile color.** Color score over time shown for each of the offspring groups (*A. loripes* purebred–orange, KL hybrid–green, and *A. kenti* purebred–blue) in each of the treatments (ambient and elevated temperature). The data points represent the mean color score and the upper and lower limits of each ribbon represent the standard error around the mean. Sample sizes (number of recruits) are included next to the data points. Color score is used here as a proxy for the density of algal symbionts in the coral tissue where a lower number/lighter color indicates a lower algal symbiont density that is indicative of coral bleaching. There was no significant effect of temperature treatment on recruit color based on linear mixed effects modelling.

standard error around the mean area of the KL hybrids, particularly for the ambient treatment, was large due to low sample sizes (Fig. 3). The LMM that included the interaction term between time point and offspring group best fit the size data and the results of this model have been presented (Table S2).

## Color

No difference in color score was detected between temperature treatments at any time point (T values −0.58 to 1.12, P values > 0.859; Fig. 4). There was no difference in color score between any of the offspring groups (T values 0.50–0.93, P values > 0.354; Fig. 4). The recruits became darker between T2 and T30 (T values = −21.74 to −20.06, P values < 0.001) and lighter between T30 and T65 (T-values = 4.47–6.06, P values < 0.001; Fig. 4). The LMM that included an interaction term between temperature treatment and time point best fit the color data and the results of this model have been presented (Table S2).

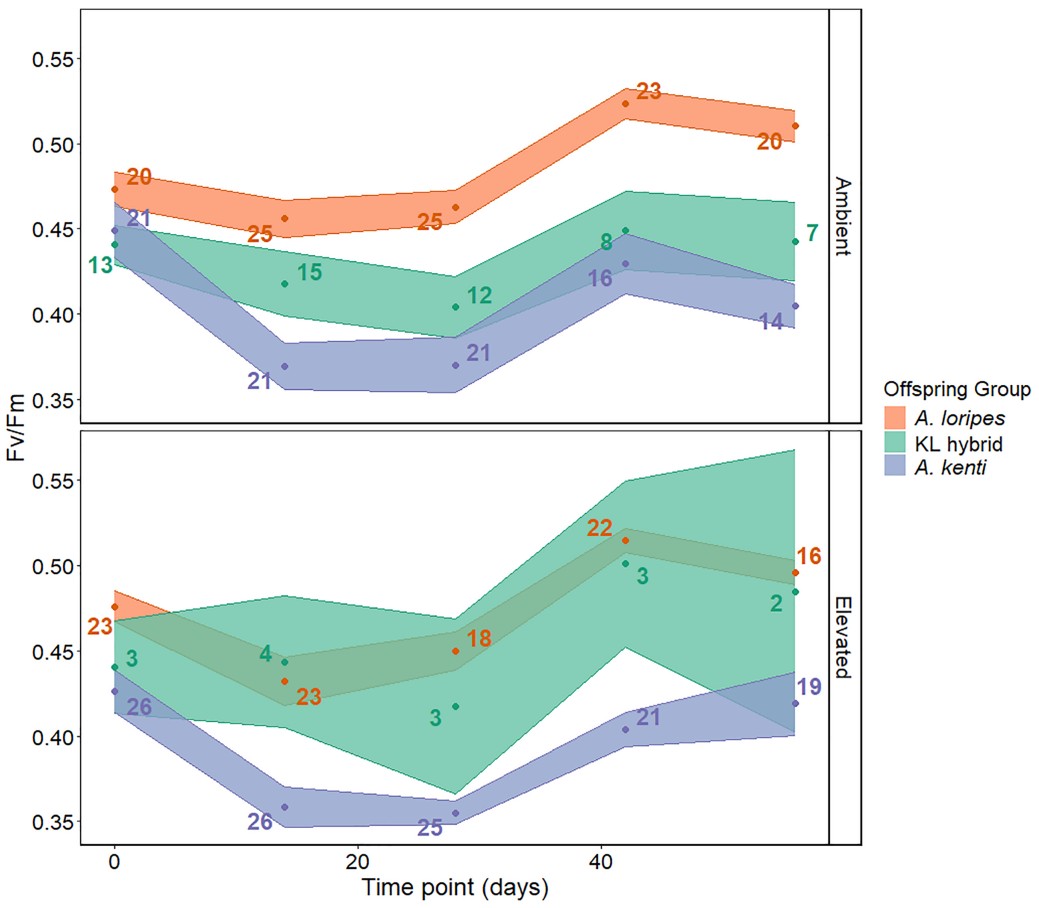

**Figure 5 Juvenile photochemical efficiency.** The change in maximum quantum yield ($F_v/F_m$) over time (days) for each treatment (ambient and elevated temperature) and offspring group: *A. loripes* purebreds (orange), KL hybrids (green), and *A. kenti* purebreds (blue). The data points represent the mean $F_v/F_m$, and the upper and lower limits of each ribbon represent the standard error around the mean. Sample sizes (number of recruits) are shown next to each data point. There was no significant effect of temperature treatment on $F_v/F_m$ based on linear mixed effects modelling.

## Photochemical efficiency

There was no significant difference in $F_v/F_m$ between the corals in the ambient and elevated treatments (T = −0.25, P = 0.815; Fig. 5), indicating that the photosystem health of the algal symbionts was not compromised by the elevated treatment. The recruits demonstrated an initial decrease in $F_v/F_m$ (Fig. 5), but $F_v/F_m$ recovered for all offspring groups and in both temperature treatments throughout the remaining course of the experiment (Fig. 5). *Acropora kenti* purebred recruits had lower $F_v/F_m$ than *A. loripes* purebred recruits (Z = −3.40, P = 0.002; Fig. 5). No significant difference in $F_v/F_m$ was detected between the KL hybrids and the *A. kenti* (Z = −1.46, P = 0.309) or *A. loripes* (Z = −1.40, P = 0.339) purebred offspring groups (Fig. 5), and the photochemical efficiency of the hybrids therefore appeared intermediate compared to the purebreds. The range of $F_v/F_m$ values recorded under ambient and elevated conditions here were similar to those observed in *A. kenti* and Symbiodiniaceae from Yunbenun under control conditions in previous

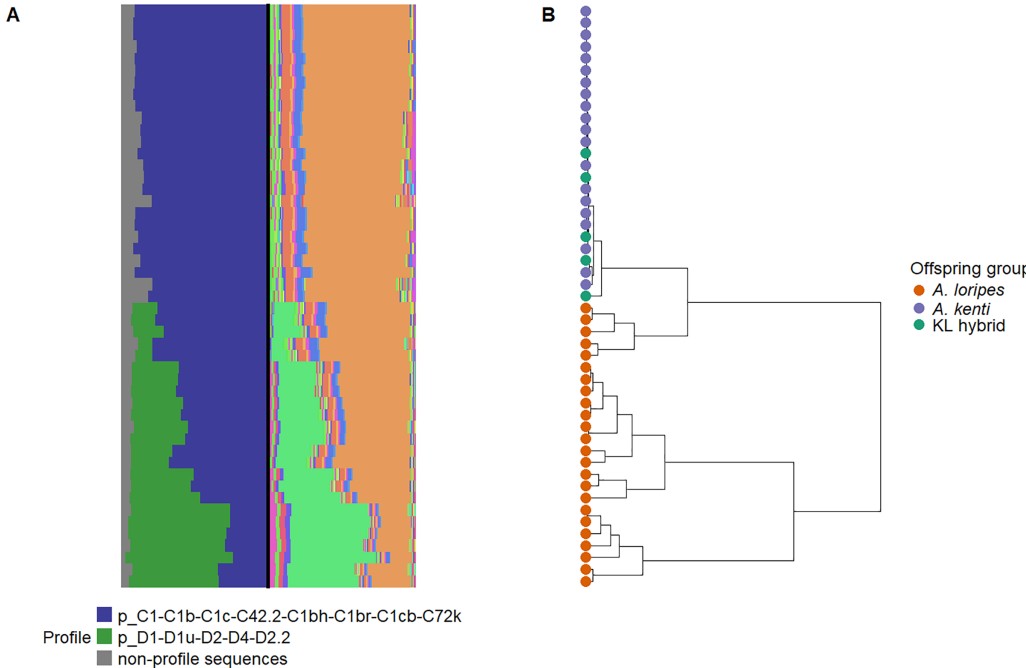

**Figure 6 Juvenile Symbiodiniaceae communities.** Symbiodiniaceae community information where each aligned row/tree branch represents data from one sample and the columns show the following: (A left) relative abundance of Symbiodiniaceae profiles that are putatively characteristic of unique taxa, where each color represents a profile, (A right) relative abundance of within-sample ITS2 sequences, where each color represents one unique sequence, and, (B) a tree that visualizes the hierarchical clustering of samples according to their unifrac distances, where the tips (representing samples) are colored by offspring group (*A. loripes* purebred–orange, *A. kenti* purebred–blue, and KL hybrid–green).

experiments (*Howells et al., 2012*; *Humanes et al., 2016*). The LMM that included an interaction term between offspring group and time point best fit the photochemical efficiency data and the results of this model have been presented (Table S2).

## Symbiodiniaceae symbiont communities

The samples had read depths between 1,327–38,448. A total of 1,071,510 reads were obtained for the 49 samples analyzed for their Symbiodiniaceae communities, averaging 29,008, 16,359 and 9,628 reads per sample from the *A. loripes*, *A. kenti*, and KL hybrid offspring groups, respectively. At the completion of the experiment, the surviving *A. loripes* corals had ITS2 profiles that were characteristic of the Symbiodiniaceae genera *Cladocopium* and *Durusdinium* (Fig. 6). In contrast, the surviving *A. kenti* and KL hybrid corals had ITS2 profiles that were characteristic of *Cladocopium* only (Fig. 6). One *Cladocopium* profile was detected amongst the samples that was characterized by the C1 majority ITS2 sequence (Fig. 6). One *Durusdinium* profile was detected in the *A. loripes* samples that was characterized by the D1 majority ITS2 sequence (Fig. 6). There was significant clustering of the offspring groups by Symbiodiniaceae community ($R^2 = 0.689$; $P = 0.001$; Fig. 6). There was no significant clustering amongst treatments by Symbiodiniaceae community ($R^2 = 0.002$; $P = 1.000$).

## DISCUSSION

### Prezygotic barriers limit interspecific hybridization

Mechanisms of reproductive isolation that prevent hybridization prior to or during fertilization (prezygotic barriers) were observed between *A. kenti* and *A. loripes*. Such barriers have been demonstrated amongst other coral species pairs (*Willis et al., 1997*; *Chan, Peplow & van Oppen, 2019*) and limit the versatility of interspecific hybridization as a restoration tool. The non-overlapping spawning times of *A. kenti* (early spawner) and *A. loripes* (late spawner) forms a prezygotic barrier that, on the reef, would restrict interbreeding of the two species. Using the approach implemented in this study, this isolation was overcome, and the results indicate that the prezygotic barrier remains semi-permeable. This is demonstrated by the unidirectional fertilization that was observed between the gametes of *A. kenti* and *A. loripes*: the eggs of *A. kenti* were successfully fertilized by the sperm of *A. loripes* while the eggs of *A. loripes* were not fertilized by the sperm of *A. kenti*. Furthermore, two of the nine *A. kenti* colonies produced eggs that could not be fertilized by the *A. loripes* sperm tested here. *A. loripes* and *A. kenti* have been previously crossed with different outcomes. *Chan et al. (2018)* successfully crossed *A. kenti* and *A. loripes* in both directions while *Chan, Peplow & van Oppen (2019)*, as per this study, were only successful in achieving fertilization in one direction; the fertilization rates between *A. kenti* eggs and *A. loripes* sperm also differed between *Chan et al. (2018)* and *Chan, Peplow & van Oppen (2019)*. Reproductive compatibility has been shown to be highly variable amongst pairs of colonies in the case of other interspecific crosses (*Willis et al., 1997*; *van Oppen et al., 2002*). Differing outcomes amongst crosses of the same species pair represent an issue for streamlining interspecific hybrid stock production. The development of genomic and/or proteomic markers of compatibility has the potential to overcome this issue. Currently, the mechanisms that determine the prezygotic compatibility between different coral species and colonies of the same species are largely unknown. Corals may have proteins on their gamete surfaces that function in a lock-and-key manner like those seen in other broadcast spawning organisms (*Vacquier & Moy, 1977*; *Ulrich et al., 1998*). Coral eggs have further been demonstrated to produce immobilization factors that initiate conspecific sperm motility (*Coll et al., 1994*; *Morita et al., 2006*). Observations of intra-specific and inter-specific variation in prezygotic compatibility may also be due to the existence of cryptic species (*Richards, Miller & Wallace, 2013*). Relatively few of the species boundaries of GBR corals have been verified using molecular techniques (*Cooke et al., 2020*; *Fuller et al., 2020*; *Meziere et al., 2024*) and as such, multiple lineages may unknowingly have been included in the crosses conducted here and elsewhere. Gamete incompatibilities may pose an issue for managed breeding of corals for restoration more broadly than just in relation to interspecific hybridization (*Miller et al., 2018*). Future work that improves the accuracy of species identification and allows prediction of breeding compatibility would help in addressing this issue.

## Hybrid coral performance was not indicative of hybrid vigor

Whilst the small sample sizes of KL hybrids limit the reliability of the results obtained here, the trends observed are largely in keeping with those seen previously for these hybrids and this adds credence to the inferences that have been drawn. Throughout this experiment, the performance of the KL recruits was intermediate (survivorship), similar to their maternal counterparts (size, $F_v/F_m$), or no different (color) compared to the *A. kenti* and *A. loripes* purebreds. Similarly, the F1 hybrid recruits of the species *Acropora florida* and *Acropora sarmentosa* did not under- or over perform relative to purebred recruits of both species in an inshore reef environment (*Lamb et al., 2024*). *Chan et al. (2018)* also compared *A. kenti* and *A. loripes* hybrid and purebred recruit performance for 1-year post-settlement; the corals were held under ambient (27 °C, 415 ppm) or elevated (28 °C, 685 ppm) temperature and $p$CO$_2$ conditions for 28 weeks post-settlement and then at ambient conditions between 28-weeks and 1-year post-settlement. The survivorship of the KL hybrids in *Chan et al. (2018)* over the first 28 weeks of the experiment was intermediate compared to their parental purebred species under elevated conditions; this is in keeping with the patterns of KL survivorship seen here under ambient and elevated conditions, noting that the temperature in the ambient treatment (27.8 °C) tanks of this experiment was closer to that in the elevated (28 °C) than the ambient (27 °C) temperature tanks in *Chan et al. (2018)*. In *Chan et al. (2018)*, the survivorship of the KL hybrids was equal to the maternal parent species, *A. kenti*, and less than the paternal parent species, *A. loripes*, under ambient conditions; after 1 year, there were no surviving *A. kenti* purebreds and fewer KL hybrids than *A. loripes* purebreds. Field deployments of coral recruits have observed hybrids to have intermediate survivorship relative to, or survivorship that is no different from their purebred counterparts (*Fogarty, 2012*; *Lamb et al., 2024*). In *Chan et al. (2018)*, the $F_v/F_m$ of the hybrid and purebreds did not differ, whilst in this study, the $F_v/F_m$ of the hybrids appeared intermediate compared to the better performing *A. loripes* and worse performing *A. kenti* recruits. The size patterns observed in this study were in keeping with those in *Chan et al. (2018)*, where the size of the KL hybrids was similar to the maternal species and was less than the paternal species over the first 28 weeks post-settlement. After 1 year post-settlement in *Chan et al. (2018)*, the KL hybrids were smaller than the *A. loripes* purebreds. The KL hybrid recruits did not display outbreeding depression/underdominance or overdominance/hybrid vigor compared to both of their purebred counterparts throughout this experiment or in *Chan et al. (2018)*. The relative performance of the hybrids compared to the purebreds under temperature stress could not be assessed in this experiment since the recruits from all offspring groups did not show any obvious signs of stress in the elevated conditions. The lack of hybrid vigor observed here indicates that the potential for hybrids to boost the resilience of reefs has limits, at least for the species pair *A. kenti* and *A. loripes*. However, the relative performance of *A. kenti* and *A. loripes* hybrids and purebreds may be different to what has been observed under tested conditions in juveniles of these offspring groups under other, ecologically relevant conditions and when they are adults. Furthermore, hybrids have performed better than

one or both of their purebred parental species based on their size and survivorship under ambient conditions, and size under elevated conditions in the laboratory (*Chan et al., 2018*), survivorship and growth on some reef zones (*Fogarty, 2012*), growth in coral nurseries (*VanWynen et al., 2021*), and resistance to disease, predation, and parasitism (*Fogarty, 2012*). Interspecific hybridization as a tool for reef restoration therefore retains merit.

## Symbiodiniaceae communities correlate with host performance

The Symbiodiniaceae communities of *A. loripes* corals differed from the *A. kenti* and KL hybrid corals in that *A. loripes* harbored *Durusdinium* and *Cladocopium* while the other groups harbored *Cladocopium* only. *Chan et al. (2019)* studied the Symbiodiniaceae communities in the same hybrid and purebred groups and, although they found no significant difference in communities amongst the groups, also observed *A. loripes* corals to harbor *Durusdinium* while the *A. kenti* and KL hybrid corals largely did not (although *Durusdinium* were detected in very low abundance in *A. kenti* growing under elevated conditions). The relative affinity of *A. kenti* to *Durusdinium* and *Cladocopium* of the C1 lineage has been shown to differ amongst life stages and environments (*Little, van Oppen & Willis, 2004*; *Abrego, van Oppen & Willis, 2009*; *Abrego, Willis & van Oppen, 2012*). Different Symbiodiniaceae communities have further been demonstrated to confer differences in coral holobiont performance, although symbionts from the same lineage can affect host performance differently depending on the environment and host species (*Rowan, 2004*; *Berkelmans & van Oppen, 2006*; *Abrego et al., 2008*; *Jones et al., 2008*; *Cunning et al., 2015*). It is thus possible that the performance differences between the offspring groups were driven by the differences in their Symbiodiniaceae communities.

## Purebred and hybrid corals demonstrated thermal resilience

The results of this study indicate that coral recruits can be resilient to the temperature stress that caused mass bleaching in adult corals (*Great Barrier Reef Marine Park Authority, 2020*; Fig. S1). A review of coral temperature experiments found that only 1% of the studies conducted have been done on recruits, signifying a critical lack of knowledge of the performance of early coral life history stages under heat stress (*McLachlan et al., 2020*). The recruits in the elevated treatment were exposed to greater than 12 DHWs relative to their parental colonies' reef of origin. Field observations have indicated that after eight DHWs, severe coral bleaching (of adult colonies) is likely and significant mortality can be expected (*National Oceanic and Atmospheric Administration Coral Reef Watch, 2021*). Significant bleaching was observed at Davies Reef during the February 2020 heatwave after approximately four DHWs (Fig. S1). Furthermore, *Hoogenboom et al. (2017)* observed significant stress in adult colonies from the genus *Acropora*, including those from the species *A. loripes* and *A. kenti*, after they had accumulated less thermal stress (~10 DHWs) on the reef than the recruits in this study experienced. These results are in keeping with a meta-analysis of studies investigating the response of coral calcification rates to temperature and acidification stress, which found that coral recruits are less sensitive to

elevated temperatures than adults (*Kornder, Riegl & Figueiredo, 2018*). Given the increasing frequency of ocean heatwaves, these results are promising both in the context of natural recruitment and out-planting of recruits in restoration efforts. However, another study which tested coral recruit performance under a more extended (70 day) 31 °C elevated treatment found survivorship to be reduced in the elevated treatment (*Quigley et al., 2020*). Coral recruits have also been demonstrated to show signs of stress under more extreme temperatures (at 32 °C; *Abrego et al., 2008*; *Yorifuji et al., 2017*). It should be noted that the results obtained here and elsewhere are specific to the experimental conditions the recruits were exposed to (*e.g.*, feeding regime and light). Field-testing the performance of recruits during bleaching events on reefs is therefore essential to estimating the breadth and degree of their resilience.

## CONCLUSIONS

Hybrid vigor was not observed in the KL hybrids relative to their purebred counterparts and this is indicative that hybrids will not always constitute coral stock with increased resilience. However, coral recruits in this study showed no obvious signs of stress under a simulation of the 2020 bleaching heatwave that occurred in the Central GBR. In reef environments where extreme temperature events are occurring more frequently than previously recorded, this is a positive sign that demonstrates a level of resilience in these *Acropora* coral recruits. However, temperature stress and bleaching has further been shown to reduce fecundity, spawning capacity, fertilization rates, larval survival, settlement, and recruit symbiont uptake, and thereby could significantly reduce natural recruitment prior to the early life stage tested here (*Negri, Marshall & Heyward, 2007*; *Abrego, Willis & van Oppen, 2012*; *Levitan et al., 2014*; *Humanes et al., 2016*).

## ACKNOWLEDGEMENTS

We acknowledge the Bindal and Wulgurukaba people who are the traditional owners of the land on which we have conducted this work. We pay our respects to their elders both past and present, and extend that respect to all Aboriginal and Torres Strait Islanders. We thank the staff of the National Sea Simulator, Katie Allen, Sophie Barkla, Wing Chan, Ashley Dungan, Heidi Hardisty, Grant Milton, Matthew Nitschke, and Britta Schaffelke for their assistance.

### Funding

This research was supported by the Paul G. Allen Family Foundation, the Australian Institute of Marine Science, an Australian Government Research Training Program Scholarship, and Australian Research Council Laureate Fellowship FL180100036. The funders had no role in study design, data collection and analysis, decision to publish, or preparation of the manuscript.

## Grant Disclosures

The following grant information was disclosed by the authors:

Paul G. Allen Family Foundation.

Australian Institute of Marine Science.

Australian Government Research Training Program Scholarship.

Australian Research Council Laureate Fellowship: FL180100036.

## Competing Interests

The authors declare that they have no competing interests.

## Author Contributions

- Annika M. Lamb conceived and designed the experiments, performed the experiments, analyzed the data, prepared figures and/or tables, authored or reviewed drafts of the article, and approved the final draft.
- Lesa M. Peplow performed the experiments, authored or reviewed drafts of the article, and approved the final draft.
- Peter L. Harrison conceived and designed the experiments, authored or reviewed drafts of the article, and approved the final draft.
- Craig A. Humphrey conceived and designed the experiments, authored or reviewed drafts of the article, and approved the final draft.
- Lorenzo Latini performed the experiments, authored or reviewed drafts of the article, and approved the final draft.
- Guy A. McCutchan performed the experiments, authored or reviewed drafts of the article, and approved the final draft.
- Madeleine J. H. van Oppen conceived and designed the experiments, performed the experiments, authored or reviewed drafts of the article, and approved the final draft.

## Field Study Permissions

The following information was supplied relating to field study approvals (*i.e.*, approving body and any reference numbers):

Great Barrier Reef Marine Park Authority.

## DNA Deposition

The following information was supplied regarding the deposition of DNA sequences:

The data and code are available at Github and Zenodo:

- https://github.com/AnnikaMLamb/Juvenile-coral-thermal-resilience
- AnnikaMLamb. (2024). AnnikaMLamb/Juvenile-coral-thermal-resilience: Lamb-etal_PeerJ (v1.0). Zenodo. https://doi.org/10.5281/zenodo.13690889.

## Data Availability

Data and code in Github repository.

Link: https://github.com/AnnikaMLamb/Juvenile-coral-thermal-resilience

DOI: 10.5281/zenodo.13690889
## Supplemental Information

Supplemental information for this article can be found online at http://dx.doi.org/10.7717/peerj.18273#supplemental-information.

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
