# Peer review of "Coral recruits demonstrate thermal resilience"

_PeerJ, doi:10.7717/peerj.18273_

## Round 0.1 · original submission · Minor Revisions

Thank you for your submission to PeerJ. Your manuscript has been reviewed by two experts in the field, and they have both provided comments which will improve the clarity of the manuscript. Please ensure that your data is included as part of your revision - it is a requirement for publication in PeerJ.

Reviewer 1 ·

Basic reporting

No comment.

Experimental design

No comment.

Validity of the findings

Since a link to the data and R scripts are not provided (as far as I can tell), I cannot be certain of the validity of the analysis. The description of the statistical methods provided appears robust, but the analyses are being run on a quite unbalanced (and in the case of the hybrids very small) sample size. I think that transparency in data analysis is increasingly important and as far as I understand is a requirement for PeerJ publications. This is an easy fix though that the authors can amend upon revision by adding a link to a data repository.

Additional comments

The manuscript is well written and structured in an organized fashion. It was an interesting read and no doubt a product of hard work by the authors. I think this manuscript would be of interest to PeerJ readers and I support its publication after some revisions. Please see specific comments below.


Title

Due to the quite low sample size of the hybrids (and the fact only TL hybrids could be produced) I think the title needs to be changed to something more general. I do not think the authors have enough data to state definitively that ‘Purebred and hybrid coral juveniles demonstrate thermal
resilience’. To be clear this is not meant to undermine the efforts by authors (making hybrids is challenging), but based on the data reported (e.g., Fv/Fm data for hybrids in the elevated temperature treatment starts at n=3 recruits; this is really not enough to make any kind of assessment of photochemical performance) I don’t think the authors have sufficient data to maintain their current title.


Abstract

Line 20: Typically ‘heatwave’ is written as a single word. From my perspective ‘heat wave’ is also fine, but this should be consistent throughout the manuscript (it is currently written both ways in parts of the manuscript).

Line 25-26: I suggest keeping the order the same when describing the egg/sperm combinations; i.e., “A. tenuis eggs were successfully fertilized by A. loripes sperm to produce ‘TL’ hybrids, but no LT hybrids could be produced from A. loripes eggs and A. tenuis sperm”.

I appreciate that it is challenging to put a great deal of detail in the abstract due to word count limits, but I think a bit more information is needed to help the reader quickly understand the study design. For example, please state the ‘level’ heat stress used (e.g., temperature or DHW) and indicate the duration of the exposure.

In the abstract (and then again in the discussion) the authors use ‘juveniles’ and throughout the manuscript they state ‘recruits’. Was this purposeful? If not, perhaps just one term and stick with it throughout.

I think the sample size limitation for hybrids needs to be included for context in the abstract for transparency.


Introduction

Line 43: I suggest changing ‘alarming’ to ‘unprecedented’

Line 54-61: In this general introduction paragraph, it is likely relevant to mention some foundational assisted evolution papers.

Line 74: I think ‘holistically’ is more commonly used. Or perhaps change to ‘comprehensive’

Line 80: Perhaps add a reference to a foundational paper that first termed ‘hybrid vigour’ here.


Methods

Line 85-86: How many colonies were collected? What were the size of the colonies? Had they experienced any heat-stress in the recent time leading up to collection that may have affected performance in the study? Is there anything unique about this reef site that readers should be aware of?

Line 94: How was this monitored, and for how long?

Line 119-122: Could the authors please expand on the rationale for this choice. Why were Symbiodiniaceae from parent colonies from Davies reef not used?

Line 122-123: Please give more information about the culture conditions – especially the temperature exposure.
Line 131-132: Did daily feeding affect the water chemistry in the tanks?

Line 152: Since the authors didn’t assess 3D images, I think the title of this subsection should simply be ‘Photo Analysis’ or something like that.

Line 153: Specify that ‘all’ tiles were imaged (as it is written it somewhat seems like only the elevated tiles were photographed).

Line 168: I suggest changing to ‘colour’ not ‘colouration’

Line 176: I don’t think the authors have defined ‘IPAM’ (though I may have missed this). Consider changing the title to ‘Photochemical Efficiency Measurements’ (or something like that) for clarity.

Line 177: I don’t think the authors need to mention Y. Those who are familiar already know this, and those who aren’t might find it confusing. What may be more useful to share in terms of specific detail (which would help future researchers) is how the authors dealt with taking PAM measurements on recruits that were quite close to one another. How did the authors ensure the adjacent recruits were still ‘dark adapted’ when they took their measurements (e.g., they weren’t blasted with light when measuring the other recruits on the tile). Could the authors expand on the approach used please.

Line 179: Be consistent with how treatments are referred to, e.g., ‘elevated treatment’ vs. heat ‘treatment’

Line 187-190: I think these lines could be deleted. Just jump right into the specifics of the analyses.


Results

Line 289: I suggest changing this to ‘Colour’ as this is how it was described earlier.

Line 295: This specification of grey value data should be stated in the methods.


Discussion

Overall, the discussion is well written, and I just have a few general comments/questions.

1. The authors discuss a lot of work by Chan, but could they discuss/compare to any other hybrid-based studies (I know there are relatively few to draw from, but this may be something to consider as it reads a little bit like a review of Chan’s papers at points). I see the authors have included some other hybrid-based references (lines 394-397), perhaps these could be drawn on a bit more in the discussion to complement the Chan references?

2. When stating ‘no hybrid vigour’ I think it is important to give some indication of whether or not the temperature tested would be enough to elicit bleaching in the field for these specific species (in lines 423-425 it wasn’t clear to me if these field observations were for the species assessed in this study or more broadly for other coral species on the source reef)…perhaps at higher temperatures the authors may have (or may not have) found evidence of hybrid vigour?

3. Do the authors think feeding the corals affected their capacity for thermal tolerance? The lab conditions may have been more favourable than that on the reef which may affect the general findings.


Figures

Figure 1: This seems like a ‘hybrid’ plot to me in some ways (i.e., a combination of real data and a schematic plot of the experimental design). I suggest choosing the ‘pure’ form of one or the other. Either show the real data with (mean with confidence interval shading), as I am sure there was variation in the ambient tanks, but this is not shown by the dashed line in the current plot. Or alternatively make this a simple schematic plot of the temperature exposure design.

Figure 2: There is a lot of blank space, and the actual data are overlapping making it hard to see clearly. Since only 2 timepoints were assessed, I think the authors could display these data more clearly to better show the data (e.g., boxplots or violin plots for the two timepoints only). Also, I know the colours are indicated in the caption, but it’s generally nice to have a legend on the figure itself too to make it fast and easy for readers to understand the results (this comment applies to other figures as well).

Figure 3,4, and 5: I personally find these plots quite busy. I suggest the authors explore another means of displaying these data (e.g.., shading to show variation and remove the boxes from the sample size perhaps? Or maybe use geom_pointrange in ggplot to help make things a little less busy?). That said, if the editor and other reviewers do not also suggest revising these figures it is fine to leave them as is (I am not seeking to drag out the review process based on plot style preferences), but just in my opinion it might be worth brainstorming other ways to display these data more clearly.

Figure 5: Fv/Fm seems quite low to me (especially for the A. tenuis); is this a typically healthy range for these species? If yes, this should be mentioned, with a reference, in the methods or results section.

General comment: I appreciate the authors being very transparent about their sample size in their plots.


I wish the authors all the best with their revision.

Cite this review as

Reviewer 2 ·

Basic reporting

This paper, which describes the larval stages of the hybrids between intercrossing coral Acropora species, presents a significant contribution to the field. The detailed observations and analyses, which are of high value, deserve to be published. I have no concerns about the experiment, from crossing design to statistical analyses.

Although the Acropora tenuis and A. loripes do not cross in nature due to their spawning time differences, the trials to make hybrids and examine hybrid vigor were conducted to understand the potential for species adaptation and selection in coral populations.

The admixture of genes between distinctive species could promote adaptation and selection at the larval stages, potentially leading to more fitted populations. However, the result showed that hybrid vigor was not observed. This phenomenon is predictable because those species are compatible but not hybridized and introgressed. Hybrid vigor could arise if two species experienced past admixtures. Although classic small data set analyses with mt data imply admixture or ILS among Acropora spp, gamete compatibility of the two relative species that speciated with changes in the spawning time might not represent the potential of hybridization, leading to the notion that hybrids do not show vigor.

The more important part of describing corals' resilience is the abundance of colonies with high genetic diversity, which leads to more fitted traits such as thermal tolerance in the new generations. Hybridization may contribute to resilience among restricted species, but its role is minor.



The following are minor points I raised before publication.

Line 356-358: "It is important to realize that gamete incompatibilities pose an...."
This is speculation, and there is no evidence presented in this study. Please weaken the statement, if possible.

Experimental design

no comment

Validity of the findings

FInding this study meets Peer J standard.

Cite this review as

---

## Round 0.2 · accepted · Accept

Thank you for being thoughtful and complete in your revision. You have taken the reviewer suggestions on board and this paper is now ready for publication. I'm excited to see it in print! Congratulations!